# Effective maternal, newborn and child health programming among Rohingya refugees in Cox's Bazar, Bangladesh: Implementation challenges and potential solutions

Malabika Sarker [1,2]☯*, Avijit Saha[1]☯, Mowtushi Matin [1‡], Saima Mehjabeen [1], Malika Asia Tamim[1], Alyssa B. Sharkey [3‡], Minjoon Kim[4], Elévanie U. Nyankesha[3], Yulia Widiati[5], A. S. M. Shahabuddin [4‡]

1 BRAC James P Grant School of Public Health, BRAC University, Dhaka, Bangladesh, 2 Heidelberg Institute of Global Health, Heidelberg, Germany, 3 Implementation Research and Delivery Science Unit, Health Section, UNICEF Headquarters, New York, NY, United States of America, 4 UNICEF, Bangladesh, Dhaka, Bangladesh, 5 Health Section, UNICEF Cox's Bazar Field Office, Cox's Bazar, Bangladesh

☯ These authors contributed equally to this work.
‡ These authors also contributed equally to this work.
* malabika@bracu.ac.bd

**Data Availability Statement:** Selected transcript quotes appear within the paper. Audio files from interviews cannot be made publicly available

## Abstract

### Background

The health status of Rohingya refugees or Forcibly Displaced Myanmar Nationals (FDMNs), especially women and children, is a significant challenge for humanitarian workers in Bangladesh. Though the Government of Bangladesh, in partnership with other organizations, is offering health care services to FDMNs, a comprehensive understanding of the program implementation is required for continuation in the future. This study explores the challenges and potential solutions for effective implementation of maternal, newborn, and child health (MNCH) programs for FDMNs residing in camps of Cox's Bazar, Bangladesh.

### Methods

We conducted a qualitative study conducted in Cox's Bazar district, Bangladesh, which involved 34 interviews (15 key informant interviews and 19 in-depth interviews) with relevant persons working in organizations responsible for MNCH services to FDMNs. We relied on both inductive and deductive coding and applied the Consolidated Framework for Implementation Research (CFIR) as a guide to our thematic analysis and presentation of qualitative data.

### Results

Our study identified some key challenges hindering the effective implementation of MNCH service delivery for the FDMNs. High turnover and poor retention of staff, overlapping of service, weak referral mechanism, complex health information system, and lack of security of the front line health providers were some of the key challenges identified. Motivating the

because of the identifying nature of these files. However, anonymised transcripts of the quotes used are available upon request to the corresponding author Prof. Malabika Sarker, email: malabika@bracu.ac.bd. The audio files and the anonymised transcripts of the quotes are part of our minimal data set The institutional review board of BRAC James P Grant School of Public Health irb-jpgsph@bracu.ac.bd The email address of non_author: Mr. Kuhel Islam The Senior Coordinator Institutional Review Board, BRAC James P Grant School of Public Health email address: kuhel@bracu.ac.bd

**Funding:** We acknowledge the support from Unicef Cox's Bazar, Yameen Majumdar, Ipsita Sutradhar, Sharmin Aktar Shitol, Chand Mia and Md Sahidur Rahman during the fieldwork in Bangladesh. We also acknnowledge USAID. Open access publication of this manuscript is supported by UNICEF through funding provided by USAID grant GHA-G-00-07-00007. The findings, interpretations and conclusions expressed in this paper are those of the authors and do not necessarily reflect the policies or views of UNICEF or USAID.

**Competing interests:** The authors have declared that no competing interests exist.

health workers, task shifting, capacity building on emergency obstetric care, training CHW & TBA on danger signs, and ensuring the security of the workers are the potential solutions suggested by the respondents. Selecting a few indicators and the introduction of E-tracker can harmonize the health information system.

## Conclusion

Providing healthcare in an emergency setting has several associated challenges. Considering the CFIR as the base for identifying different challenges and their potential solutions at a different level of the program can prove to be an excellent asset for the program implementers in designing their plans. Two additional domains, context, and security should be included in the CFIR framework for any humanitarian settings.

## Introduction

Globally, by 2018 conflicts have displaced 70.8 million people that includes 13.8 million newly displaced, and since 2009, the gradual increase in humanitarian crises is now a global concern [1]. The world witnessed one such crisis in August 2017, when Rohingyas were forcibly displaced from neighboring Myanmar; about 706,364 Rohingyas have cumulatively arrived in Bangladesh since then [2, 3]. As a non- signatory to the 1951 Refugee Convention, the Bangladesh Government officially refers to them as 'Forcibly Displaced Myanmar Nationals' or FDMNs [4]. The recent displacement, coupled with a previous influx, has created the world's most densely populated FDMN settlement in Cox's Bazar with an estimated 911,566 Rohingyas currently living in different camps [5–7].

In total, five campsites situated in Ukhia and Teknaf Subdistrict of Cox's Bazar, Ukhia shelters more than two-third of the total refugee population (Fig 1). The refugees living in these settlements are sufferers of hunger, nutrition, safety, and other medical emergencies. The last Joint Response Plan (JRP 2019), identified that more than one million FDMNs were in need of different health-related services [8]. In the FDMN camps, 52 percent are female, of which 23 percent are between the ages of 18 and 59 years [9]. Following the displacement, 474,014 women and girls are living in the camps situated in Cox's Bazar district of Bangladesh, among them, about 22,000 were pregnant as of January 2019 [8].

Together with the government of Bangladesh, more than one hundred national non-governmental organizations (NGOs), international NGOs, United Nations (UN) organizations, and several donor agencies have been providing both preventive and clinical care, including health promotion, for the FDMNs since the start of the influx. Maternal, Newborn, and Child Health (MNCH) services are the primary focus of the interventions [11]. The FDMNs receive healthcare services from the primary health centers and health posts located within the camps. More than 200 doctors and nurses are currently providing services in the health centers and health posts. These health centers and health posts offer a variety of MNCH services, including Antenatal Care (ANC), Post Natal Care (PNC), referral, normal deliveries, and counseling on IYCF indicators. Apart from the facility-based services, more than 1200 community health workers are providing community-based counseling and other services and assisting in referral to different facilities [12].

The service package of sexual and reproductive health (SRH) and access to essential reproductive, maternal and newborn health services remains a significant concern. Recent analyses

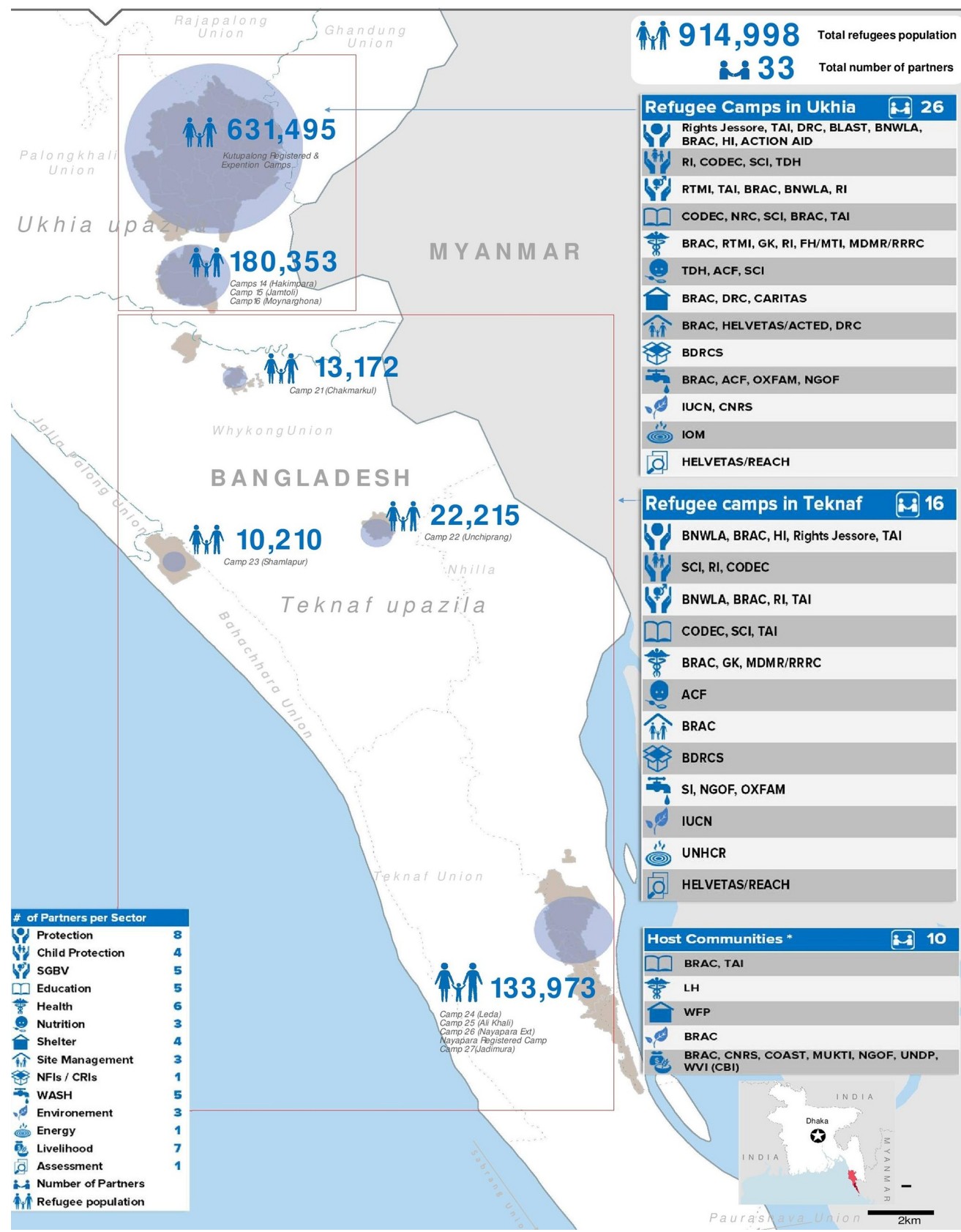

**Fig 1. FDMN campsites with population and major stakeholders [10].**

reveal that overall coverage of MNCH services is yet inadequate for a variety of reasons [13–15]. The majority of FDMN women give birth at home: recent analyses suggest that only one out of five pregnant mothers sought delivery care from the health facilities inside the camps for varying reasons, including the prohibition from the family and distrust about the facility-based services [16,17]. Further, between September 2017 and August 2018, 52 maternal deaths out of 82 pregnancy-related death occurred within these camps [18].

Even after two years of the major influx, the repatriation of the FDMNs is still unpredictable. Until that happens, the continuation of MNCH service delivery is necessary, and organizations currently working for FDMNs are accountable for continuing their services. The challenges in implementing a diverse spectrum of activities in an emergency setting are irrefutable. Implementation challenges are further accentuated by the tension between the host community and one of the largest FDMN camps on their doorstep [19]. Addressing ongoing implementation challenges of MNCH programs and adopting solutions to meet the critical needs of the population are pivotal. However, little has been done to create a scientific evidence base of the challenges that organizations are facing during their service delivery nor of strategies to address them. The research questions of this study were framed in collaboration with the relevant stakeholders in a consultative workshop. This paper aims to present a comprehensive understanding of the existing implementation challenges of MNCH programs among Rohingya refugees as well as solutions that can help policymakers to reshape and streamline assistance for FDMNs until repatriation occurs.

## Methodology

### Study design

The study relied on a qualitative approach, and we conducted in-depth interviews (IDIs) and key informant interviews (KIIs) with respondents from different organizations working on MNCH service delivery to FDMNs in Cox's Bazar, Bangladesh.

### Study setting

This study was conducted in Cox's Bazar district, which is located in the southeast part of Bangladesh. The study specifically investigated the MNCH program implemented in the Kutupalong refugee camp in Ukhia Subdistrict, Cox Bazar. The camp in Ukhia alone accommodates 734,645 of the total FDMN population, and currently, 27 organizations, including UNHCR, are providing varieties of services in those camps (Fig 2). The camp also harbors primary, secondary, and referral centers accountable for MNCH care.

### Study duration and study population

The study was conducted between April 2019 and September 2019. Study participants included a range of stakeholders including MNCH program managers working local NGOs, international NGOs, UN organizations, and service providers in health posts and primary health care centers (i.e., doctors, nurses) run by public and private sectors and community health workers involved in providing MNCH services to the FDMN community.

### Data collection

A series of IDIs and KIIs with purposively selected participants were conducted using pre-tested semi-structured interview guides. We sampled our study participants purposively. We conducted the KIIs with upper mid-level and senior managers working at different levels in different major organizations (GO, NGO, INGO, UN). They are well acquainted with national

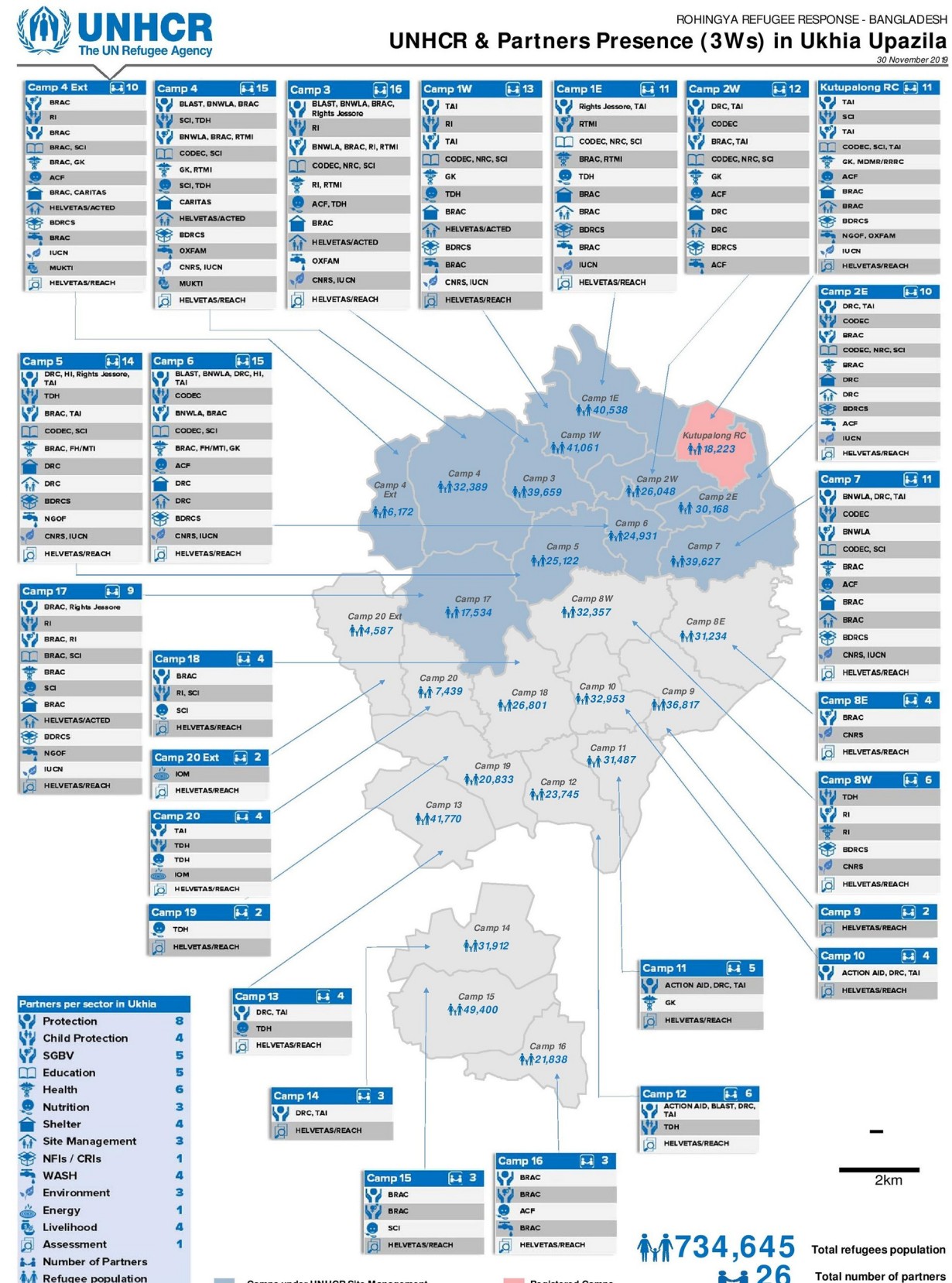

**Fig 2. FDMN campsite and stakeholders presented in Ukhia subdistrict [10].**

**CFIR Construct: Implementation Process**
- Challanges
  - Service Delivery
  - Challenges to Referral services
- Facilitators
  - Increase Coordination among organizations
  - Strengthening Referral

**CFIR Construct: Inner Setting**
- Challenges
  - Human Resource: Frequent Turnover of Staff
  - Human Resource: Capacity Development of New Recruits
  - Human Resource: Necessity of Particular Cadre
  - Supply Chain Management
  - Health Information System
- Facilitators
  - Human Resource: Staff motivation and Capacity Building
  - Central coordination and long-term agreements for an effective supply chain

**CFIR Construct: Outer Setting**
- Challenges
  - Transportation Channel for delivering supplies to the camps
  - Shortage and Cutting Down on Funds

**CFIR Construct: Unique for Emergency Setting**
- Context of the Situation
- Security of the staff

*The CFIR Constructs "Characteristics of the Intervention" and "Characteristics of the Individuals" were not applicable for this study

**Fig 3. CFIR constructs and overlaid themes with associated challenges and facilitators.**

**Table 1. Respondents' profile.**

| Type of Organization | Designation of the Participants | Type of interviews conducted | Number of interviews conducted |
|---|---|---|---|
| Government | Medical Officer Civil Surgeon | KII | 1 |
| InternationalNGO(INGO) | Head of Program support | | 1 |
| | SRH Sector Lead | | 1 |
| | National Program Officer | | 1 |
| | Program Manager | | 1 |
| | Field Health Sector Coordinator | | 1 |
| | Health Manager | | 1 |
| | Medical Coordinator | | 1 |
| UN | Midwifery Specialist | | 1 |
| | Public Health Officer | | 1 |
| Local NGO(LNGO) | Medical Officer | | 3 |
| | Clinic In-charge | | 1 |
| | Team lead | | 1 |
| INGO | HMIS Manager | IDI | 1 |
| UN | HMIS Officer | | 2 |
| | SCM Manager | | 1 |
| | C4D Specialist | | 1 |
| LNGO | Paramedic/Midwife | | 4 |
| | CHW | | 8 |
| | Majhi (CHWs from FDMN community) | | 2 |

policies, implementation strategies, and delivery mechanisms. The IDIs respondents were health service providers working for relevant organizations and responsible for providing health services to the camps. In total, we conducted 34 interviews (15 KIIs and 19 IDIs). We asked the participants about MNCH challenges and potential solutions circumnavigating the implementation of the program. We contacted the participants beforehand to ensure their availability, with the help of Unicef Cox's Bazar, our local partner, for the study. We pre-tested the IDI & KII guidelines in a similar facility, which was not part of the primary research before the data collection. A seven-member research team experienced in conducting qualitative interviews took part as the interviewers, all of whom had received training before going to the field to get familiar with the guideline and study objectives. We carried out the interview sessions in a setting free from outside interference so that the study participants could feel at ease in sharing their views and perceptions. We conducted our interviews until there was no new information generating from the interviews. We obtained written informed consent from each of the participants before the interview as we recorded the interviews using digital audio recording devices. The Institutional Review Board (IRB) of BRAC James P Grant School of Public Health granted the ethical permission for the study. Permission to conduct this research in the FDMN settlement areas was obtained from the office of Refugee Relief and Repatriation Commission (RRRC) of the Bangladesh government. Staff from the UNICEF Cox's Bazar field office provided support to the team to access the camps and to identify the most relevant respondents for this research.

## Data analysis

A three-member team of mid-level researchers took part in familiarizing and organization of relevant codes in a way to represent the participants' views and to reduce potential biases. The descriptive qualitative method [20] guides analysis by describing the data in common terms

and relating to the area of study. We also followed the five-step framework analysis [21,22]. First, the research team reviewed transcripts and audio recordings for familiarity. Following that, in the second and third steps, codes and themes emerged based on the inductive and deductive coding. Then the codes were indexed into relevant 'Consolidated Framework for Implementation Research' or CFIR domains (Fig 3), and the new emerging codes were assigned to new domains. CFIR guides the systematic assessment of implementation contexts to find out factors that influence intervention implementation and effectiveness [23]. We organized and charted the codes under themes as the fourth step to better reflect the participants' responses. In the final step, we reviewed the codes and associated themes multiple times to ensure the inclusion of the participants' words, to improve the credibility of the interpretation. To avoid any sort of intra-code discrepancy among the researchers who took part in the analysis, we cross-checked and counter-checked all the codes that emerged and organized the themes when we reached a consensus. Nvivo 10 software was used for the organization and management of the data.

# Results

## Participant characteristics

In total, we conducted fifteen KIIs and nineteen IDIs (Table 1).

## Challenges and potential solutions to MNCH service delivery among FDMNs

The study identified a number of challenging factors and potential solutions related to MNCH service delivery. We presented the challenges following the three domains of the CFIR framework: 1) 1) Implementation process 2) Inner Setting, 3) Outer Setting and with two additional domains, 4) Context' and 5) Security relevant to the humanitarian crisis. The relevant sub-themes are presented within each domain.

## Challenges

**Context.**   FDMNs' condition within the settlements of Ukhia and Teknaf Upazilas (sub-districts) has improved significantly since the arrival of the refugees in 2017. Many temporary shelters in the form of tents have been transformed into housing, separated by pathways three-feet wide and a few roads that are hard to use. Soil erosion is a growing problem, and the risk of deadly landslides triggered by heavy rains is a significant barrier to access health care services.

**Security.**   Due to the uniqueness of service delivery in an emergency setting, the safety of the service providers is the biggest constraint for the effective implementation of the MNCH programs. Field staff reported that they threatened by the very beneficiaries they were serving. FDMN community members sometimes verbally assault the service providers if they cannot deliver medicine. Moreover, health care providers like doctors and nurses, feel unsafe while working at health posts, particularly without the protection of secure fences, despite the presence of guards at the health facilities.

*"Why*? *This Rohingya community is very violent. If there is any dispute over some issues, they become very dangerous. You really cannot tell what will happen when you are going to work or walking on the road." Res 01_PI_LNGO*

## Implementation process

**Service delivery.** Many organizations offer MNCH services inside the FDMN camps. The overlapping of services is a big challenge and often creates a problem, such as targeting the beneficiaries and the allocation of funds appropriately.

*"There have been some areas where services are overlapping, and underutilization of resources is happening in other facilities. Different organizations for the same beneficiaries deliver similar activities." Res 24_PI_INGO*

The majority of the centers do not offer 24/7 services, and the laboratory services are very limited. However, in health posts, blood grouping and detection of urine albumin are possible.

**Referral.** The referral mechanism in the camp setting is fragile and unstructured. The patients usually are referred to the only referral health facility within the camp, which provides comprehensive emergency obstetric care and the government-run sub-district level health complexes, which provides secondary level health services to the patients. The poor coordination and lack of capacity are the main underlying reasons for the weak referral system. These are exacerbated due to the nonexistent referral guideline, shortage of referral facilities, absence of referral manager, the distance of the referral facilities, and lack of 24/7 transportation facility.

*"Yes, it is a big challenge as the referral hospital might be a bit distant. who will bring her to the facility, bear the cost, she also needs one attendant." Res 32_PI_INGO*

Sometimes, the health providers at referral facilities avoid receiving complicated patients due to the shortage of specialists or high patient load.

*"The lack of capacity of the referral health facility is a problem. Upazila Health Complex is a 50 bedded hospital. Sometimes the patient load is high. Therefore, they provide initial treatment and send the FDMN clients back home." Res 02_PI_LNGO*

Providing MNCH services inside the FDMN camps require unified efforts from the field level staff and the facility level staff. However, due to the presence of untrained traditional birth attendants from the FDMN community and sometimes inexperienced midwives or nurses, handling of emergency cases become complicated. The inexperienced midwives often lack confidence and fail to refer cases on time.

*"How would they (TBAs) know? If the case (pregnant mother) was easy, then they may ask the 'Khalash' (to perform the delivery). In case of any problem, the TBA would never know that. It is what we try to make them (pregnant women and their guardians) understand." Res 20_CO_LNGO*

## The inner setting

Providing MNCH services inside the FDMN camps requires unified efforts between the staff who are working at the community level and at the health facilities inside the camp.

## Human resource

**Frequent turnover of staff.** The frequent turnover often hampers the MNCH service provision. Many respondents (6) mentioned that the retention rate of health providers in the

health posts inside the FDMN camp, especially doctors, nurses, is very low. One of the main reasons is a better opportunity for young doctors for career building. It results in the recruitment of new and often inexperienced staff and consequently a decline in the quality of the service.

> *"For example, many doctors appear for BCS (Bangladesh Civil Service) exam and will leave if selected. So a new batch will come, and if they find a better offer elsewhere, will leave too." Res 02_PI_LNGO*

The frequent turnover is also typical among other staff, especially MIS officers. The loss of experienced and trained personnel at different levels contributes to poor quality of care and additional use of resources and funds for the capacity development of recruits. Despite the frequent turnover, vacant posts are filled up almost immediately. Internal recruitment was preferable to avoid the pressure from local political and influential leaders to recruit their candidates.

**Capacity development of recruits.** The recruits require immediate training to boost their knowledge and skills. However, few respondents mentioned attending training not relevant to their works was another challenge. While organizing new training is common, refresher training or post-training supervision is usually rare. Attending Training also causes a problem such as the disruption of the service provision.

> *"Sometimes, there is a request for multiple training on the same day. And if the doctor leaves for training, then the facility is empty." Res 39_PI_UN*

**The necessity of particular cadre.** Many of the respondents (7) mentioned that the shortage of staff is not a problem unless staff goes for a holiday or there is a sudden surge of patients. However, rational distribution and availability of specialized staff are still missing.

> *"Why should I have ten health facilities when I do not have enough staff to run all of them? I would rather have few do it perfectly than to have so many. . . . . . . . . . Human resource is not rationally distributed. Some have more some have less. On the side of MNCH, I think most organizations do not have enough midwives." Res 41_PI_INGO*

The health posts require more female doctors and midwives, as the conservative FDMN community preferred them as service providers. Additionally, three respondents mentioned appointing more field monitoring staff can improve the quality of service through tracking and monitoring.

**Supply chain management.** The maintenance of medical supplies poses concern as all warehouses don't have qualified logistician and thus fail to maintain quality. Many of the medicines are temperature sensitive and need to be preserved in the storehouse with a functional cooling system or air-condition, which is missing in many stores. There is also a shortage of anesthetic agents for manual removal of the placenta and Vacuum extractor for normal delivery.

The injectable form of both painkillers and antibiotics for babies, including pediatric dropper for liquid medicine for neonates are also in shortage. Scarcities and delay in the refilling of iron and folic acid, vitamin, calcium tablets are noted.

> *"As most of the patients are children, the scarcity of pediatric medicines is common. We need them most. We try to manage as much as we can. But if the gap remains, we have nothing to do." Res 01_PI_LNGO*

**International procurement consumes more time.** Almost all international organizations procure their medical supplies from abroad due to their unavailability in the local market and to maintain a good standard. They do so by having long term agreements with their suppliers. However, this takes up too much time for the steps involved in the process. One respondent said,

*"We have international procurement, which is very time-consuming. . . . It comes from abroad in a shipment, and we need a shipment clearance and stuff. We follow the normal procedures and systems, but clearing the shipment is difficult." Res 39_PI_UN*

**Health information system.** The challenges related to the health information system is not uncommon. All respondents mentioned that report to the common DHIS-2 platform set up by the Government is mandatory. According to one respondent, complexity arises because of too many indicators and frequent changes in the format of the report based on an individual's choice. Inconsistency of data uploaded on the website and over-reporting are other significant challenges with the health information system. The over-reporting is common due to the absence of a tracking system, lack of coordination among the different organization, and weak monitoring system.

*"There are too many indicators, which should be reduced. Whenever someone comes, that thinks of adding something here, right here. Formats are randomly created whenever someone needs it." Res 24_PI_INGO*

Several respondents mentioned the absence of an individual patient tracking system makes it hard to provide a continuum of care. The inaccuracy of the data due to the mistakes during medical diagnosis made by the data entry operator with no exposure to medical vocabulary has been reported frequently. Variety in the types of reporting and a large number of report submissions is quite challenging for the respondents. The differences among the organizational priorities often hinder data sharing opportunities. In the respondent's voice,

*"Different organizations have different reporting and requirements, and it is hard to fulfill all of them." Res 39_PI_UN*

## Outer setting

**Transportation channel for delivering supplies to the camps.** Providing supplies to the beneficiaries is problematic due to the lack of proper transportation channels. The poor network of roads, especially damaged roads, making it hard to transport supplies without damaging the supply. Some organizations rent vehicles for transporting supplies, which are time-consuming and incurs extra costs.

Also, there is a big challenge for transporting supplies to the warehouses located in the camps. FDMN community sometimes blocks the truck movement because they are afraid of the accident for themselves and for the children who use the road as a playground.

**Shortage and cutting down in funding.** Implementing the MNCH program is becoming difficult not only because of the gradual shrinkage of the funding but also due to the pressure from the donor agencies to cut down the expenditure. Organizations can rarely employ international staff qualified for working in such an emergency setting because of the lack of funds. The human resources available in Bangladesh, especially for the higher-level positions, are not

adequately qualified for working in a humanitarian crisis setting. Retaining highly qualified Bangladeshi personnel requires higher salaries. Moreover, funding is also needed to provide continuous training and capacity building for all the staff.

## Solutions

**Security.** Another challenge against an enabling environment is a gradual increase in crime and violence. The extensive and comprehensive coordination and cooperation between the government, the army, and the implementing organizations have been launched to address the security concern.

*"Bangladesh Army has played a great role. They ensure road connectivity of some strategic location and also traffic control within the camp, security management as well. Along with the Army, Bangladesh Police force also took the visible initiative, which facilitates the task of supply change management easier, and we will progress day by day." Res 23_PI_INGO*

Besides that, UN agencies have policies and guidelines to ensure the security and wellbeing of humanitarian responders. Any unwanted incidence is managed according to the protocol. The *'Majhi'* is a community leader in the FDMN population and appointed by the different organizations to act as community-level volunteers for the social mobilization activities, often helps other CHWs in their works by using their influence as the local community leaders.

*"Block "Majhis" are prior informed [from the hospital], for solving any problem. However, while walking, some people comment on us, and Then "Majhi" tell the community look they are like us and respect them like your mother and sisters." Res 20_CO_LNGO*

## Service delivery

**Increase coordination among organizations.** Three respondents mentioned that coordinated efforts between the government and other implementing organizations to reduce overlapping of service delivery are ongoing but not fully functional. Some overlaps are unavoidable, but through mapping of the working areas with allocating the activities can reduce further overlapping.

Another coordinated effort is task-shifting at the field level that reduced the high workload at the health facilities. Some health facilities are already using to tackle higher patient load.

**Strengthening referral.** An effective referral system is critical for ensuring timely care. A standard solution to overcome the barriers to referral is to employ and train staff, especially the referral managers, and the creation of a rotational duty. Forming a partnership with the private facilities and better communication with the existing referral clinics in the camps is critical for an effective referral system.

*"When the patient is in an emergency, we communicated with the private obstetrician. They treat or operate the patient. We reimburse the clinic afterward." Res 30_PI_INGO*

## Human resource

**Staff motivation.** Motivating the overburdened staff is another effective strategy. The provision of performance-based incentives, both in-kind and cash, has been introduced. Many organizations select the best performers and send them to specialized training or special events

both within and outside Bangladesh. Other strategies include supportive supervision, delivering an inspiring speech in the meeting, and provision of health insurance.

*"What we do is specific and unique training, and we cannot send all of them. We want to send someone who is committed, hardworking, collaborative, and energetic, that's whom we prioritize for those specific and important training. . . That's one way of motivation. Second, we promoted several staff for their performance." Res 40_PI_INGO*

**Capacity building.**   The standard strategy adopted by the organizations to tackle the shortage of experienced staff is the provision of in-house training sessions with regular refresher training. The working teams provide each other support and guidance daily, sometimes even over the phone. The monthly or weekly staff meetings are a platform for discussing the gaps and the solution.

*"Yes, those who are newcomers, they might not be knowledgeable about everything. They are continuously learning, and the older ones are helping them with it. We are giving them guidelines and guidance. Problems are also being solved almost immediately by arranging formal training and meetings." Res 23_PI_INGO*

The organizations maintain staff capacity assessments using checklists and tools to identify training needs. They organize weekly continued medical education sessions and other knowledge-sharing sessions. In the morning sessions, doctors and supervisors discuss with the colleagues about ANC, PNC, breastfeeding, etc. Sometimes role-play of counseling sessions is conducted to practice how a counselor can convince a pregnant woman to visit the facility. Various external training was also organized by different agencies like UNICEF and others in collaboration with the government.

As there is a shortage of MIS staff, other staff are trained for the data reporting and related tasks and to use tablets instead of doing tally sheets on paper.

The logisticians received supply chain management training for ensuring quality control. One respondent said,

*"When they detect they have a problem, they call on me, or they write me an email (saying things like) "I need support. Can you send me a team to come and help me organize my warehouse" I dispatch two people, (saying) "Go to location X, and fix that warehouse? Put it to the standard"" Res 27_PI_UN*

Expiry dates are monitored using color-coding the stocks in a spreadsheet for identifying the shelf life of the medicines. The warning system allows the providers to consume the drugs before expiration. A guideline has been developed to dispose of expired medication.

*"So we have a guideline on managing expired drugs. So I think, last year it has been developed, and it is also work in progress." Res 38_PI_INGO*

## Supply chain management

**Central coordination and long-term agreements for an effective supply chain.**   Previously when the supply of medicines and other logistics were in dire need, the regular procurement process was made flexible by using a longstanding agreement with the supplier.

Therefore, the procedure was easier and quicker, but it became slower. In this respect one respondent said,

*"At the beginning of this emergency, receiving supplies required one week or 15 days if you have a long term agreement with the supplier. It took only three days.." Res 27_PI_UN*

**Validating local procurement sources.**  Some implementers procured medicines and logistics in greater quantity and ordering it twice a year rather than three times a year. The gradual increase of validated, locally manufactured medications can reduce the burden of international procurement and save time. One respondent in this regard said,

*". . . . . ..Bangladesh has many pharmaceutical companies. . . . . . . ..while waiting for prequali-fication from WHO, each organization can also do the quality standard by in-house quality team. We already have data that are validated. The number of drugs for local purchase will continue sometime soon." Res 41_PI_INGO*

## Health information system

Some respondents (2) suggested a unified reporting system like the DHIS-2 platform. It can reduce redundancy in data and the workload of the staff. One HIMS manager empha-sized on disaggregation of data and the introduction of an e-tracker system to improve data quality along with the tracking of beneficiaries. As harmonizing of the data collected is always a big challenge, some respondents prefer to select a few relevant indicators. They believed proper coordination among different organizations might help in doing all those easily.

*"So there is a need of harmonizing data system and see that what needs to be collected, what you are using that for. So, that's some health sector taking priority to see that how we can har-monize the system." Res 38_PI_INGO*

## Discussion

This study tried to demonstrate the challenges related to MNCH service delivery in an emer-gency setting like the crisis concerning the FDMNs and some potential solutions to those chal-lenges following a unique framework for implementation science, the CFIR. The CFIR guided us for interpretative coding and creating the challenges and possible solutions into groups for further clarification of the issues related to program delivery in a humanitarian crisis setting [24].

At the intervention level, this study identified several barriers across the themes. The most prominent ones are the frequent turn over of staff, weak referral system, the complexity in analysis and reporting of data, difficulties in procurement, and security concerns. High turn-over and poor retention of health staff, especially for field-level, has always been a problem in the FDMN context [25–27]. By making a rational distribution of healthcare staff and address-ing the challenges related to the cultural characteristic of the FDMNs can earn great benefits for the organizations working in Cox's Bazar. Previous reports indicated the same setting indi-cated problems regarding the unequal distribution of healthcare staff, and poor-quality ser-vices rendered along with poor behavior shown by the staff [25, 28]. These reports also

mentioned extra burden for providers due to the heavy workload and unequal distribution in different areas. The effective patient referral system is a significant concern in developing countries [29]. Without a functional referral system, it is not possible to provide comprehensive health care to pregnant women and children on time [26]. Previous studies done in the FDMN camps [30,31] showed that the referral pathway was too convoluted, and there was also underutilization of referral services by the beneficiaries. Our study found that the situation has been improved after the introduction of dedicated vehicle service and referral managers. An effective referral system is an indicator of a functional health system and can be a data source to understand how the community perceives the care they are receiving [32,33]. Strengthening the role of CHWs and empowered community engagement can be a key to improving the referral system [34]. Our study revealed that training midwives on basic emergency obstetric care in selected facilities would avoid serious complications and will ensure timely referral. Family-based counseling at the community level by the local CHWs, training TBAs, and CHW on pregnancy-related danger signs and motivating TBAs for early referral are the possible solution for strengthening the referral system. Supportive supervision with the recruitment of experienced midwives in the referral facility will facilitate the admission process.

Another intricate finding from our study suggested that the program implementers should consider a lack of safety associated with service delivery inside the FDMN camps. With thousands of unemployed young men with an uncertain future ahead, it is not surprising that some of them will resort to crime as trafficking. Also, drug rings are notoriously active in these camps. The women and young girls are the most vulnerable and often become the victim of the violence. While the personal security of the displaced people or refugees has been emphasized before [35], however, the security issues faced by the field level workers have hardly been recognized. The security measures must be influenced by the legal protection of both refugees and the aid workers. As unique as a humanitarian setting can be, a risk assessment for the healthcare staff, and careful planning and monitoring mechanism should be considered while planning for service delivery. In such a situation, providing a secure and better environment can ensure better service delivery.

The poor record system of the information provides heterogeneous information about FDMN. Some studies conducted among the FDMNs in recent times found traces of such differences [36,37]. Creating a unified reporting system can improve the record-keeping system. Harmonized coordination among different stakeholders is essential to execute health programs, particularly in a humanitarian setting [37,38]. The strong links between international and national partners are crucial to achieving a more effective response and promoting medium-term recovery and building resilience to shocks.

Finally, the donors and program implementers need to develop a well-coordinated strategic plan for their next step. Building capacity of the mid-level managers and health care workers, together with a functional referral system and an effective supply chain management, should be considered.

The study has a few limitations. Due to the resource constraints, the IDIs were limited to the service providers in the largest camp. Although this camp hosts more than 70 percent of the FDMN population, variation in the challenge's providers working in the other camps' experience was not captured. Due to the time limitation, the CFIR domain individual characteristics and characteristics of the intervention were not investigated.

In conclusion, the CFIR can prove to be a great asset in helping the implementers and donors in evaluating and designing their future strategies. Future research is highly encouraged to explore and address the challenges related to other types of service delivery in a humanitarian crisis setting using the CFIR.

## Author Contributions

**Conceptualization:** Malabika Sarker.

**Data curation:** Avijit Saha, Saima Mehjabeen, Malika Asia Tamim.

**Formal analysis:** Malabika Sarker, Avijit Saha.

**Funding acquisition:** Malabika Sarker, A. S. M. Shahabuddin.

**Methodology:** Malabika Sarker, Alyssa B. Sharkey, Minjoon Kim, Yulia Widiati, A. S. M. Shahabuddin.

**Project administration:** Malabika Sarker, Avijit Saha, Mowtushi Matin.

**Resources:** Minjoon Kim.

**Supervision:** Malabika Sarker, Avijit Saha, A. S. M. Shahabuddin.

**Visualization:** Avijit Saha.

**Writing – original draft:** Malabika Sarker, Avijit Saha, Mowtushi Matin, Saima Mehjabeen, Malika Asia Tamim.

**Writing – review & editing:** Malabika Sarker, Avijit Saha, Mowtushi Matin, Alyssa B. Sharkey, Minjoon Kim, Elévanie U. Nyankesha, Yulia Widiati, A. S. M. Shahabuddin.

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
