## [Decision Letter · Decision Letter 0]

22 Nov 2019

PONE-D-19-28648

Effective maternal, newborn and child health programming among Rohingya refugees in Cox’s Bazar, Bangladesh: Implementation challenges and potential solutions

PLOS ONE

Dear Dr. Sarker,

Thank you for submitting your manuscript to PLOS ONE. After careful consideration, we feel that it has merit but does not fully meet PLOS ONE’s publication criteria as it currently stands. Therefore, we invite you to submit a revised version of the manuscript that addresses the points raised during the review process.

We would appreciate receiving your revised manuscript by Jan 06 2020 11:59PM. To enhance the reproducibility of your results, we recommend that if applicable you deposit your laboratory protocols in protocols.io, where a protocol can be assigned its own identifier (DOI) such that it can be cited independently in the future. For instructions see: http://journals.plos.org/plosone/s/submission-guidelines#loc-laboratory-protocols

We look forward to receiving your revised manuscript.

Kind regards,

Vijayaprasad Gopichandran

Academic Editor

PLOS ONE

Journal Requirements:

2. Please include additional information regarding the survey or questionnaire used in the study and ensure that you have provided sufficient details that others could replicate the analyses. For instance, if you developed a questionnaire as part of this study and it is not under a copyright more restrictive than CC-BY, please include a copy, in both the original language and English, as Supporting Information. In addition, please state upon how many participants the survey was pre-tested.

4. Please ensure you have thoroughly discussed any potential limitations of this study within the Discussion section.

Reviewers' comments:

Reviewer's Responses to Questions

**Comments to the Author**

1. Is the manuscript technically sound, and do the data support the conclusions?

Reviewer #1: Yes

Reviewer #2: Yes

2. Has the statistical analysis been performed appropriately and rigorously? 

Reviewer #1: N/A

Reviewer #2: N/A

3. Have the authors made all data underlying the findings in their manuscript fully available?

Reviewer #1: Yes

Reviewer #2: Yes

4. Is the manuscript presented in an intelligible fashion and written in standard English?

Reviewer #1: Yes

Reviewer #2: Yes

5. Review Comments to the Author

Reviewer #1: Title: Effective maternal, newborn and child health programming among Rohingya refugees in Cox’s Bazar, Bangladesh: Implementation challenges and potential solutions

Version: 1.

Date: 13 November, 2019.

The paper developed and illustrated very nicely! I have no major comments:

Minor Comments:

• You have conducted 15 KIIs and 19 IDIs, but what was the difference between IDI and KII regard to objectives? In your sample table how you have distinguish the IDI and KII participants? Were there different interview guidelines for two different groups? Objective(s) was different? Findings were different? Clarifying these issues could strengthen the article.

• Author has mentioned that they conducted 15KIIs and 19 IDIs, but did not mention that why it was 15/19? Data saturation reached or not or any constraints in proceeding for more.

Declaration of competing interests: 'I declare that I have no competing interests'.

Reviewer #2: Thank you for the opportunity to review the paper entitled, “Effective maternal, newborn, and child health programming among Rohingya refugees in Cox’ Bazar, Bangladesh: Implementation challenges and potential solutions.” The authors are to be commended for working on such a revealing and well-documented study in the challenging environment of a humanitarian crisis. The qualitative study of 24 in-depth interviews and key informant interviews provides insight into the challenges and suggested solutions for improving maternal, newborn and child services in the Rohingya camps. I believe this manuscript will add value to the literature around humanitarian and fragile settings. There are some areas for consideration and refinement.

Major:

• As PLoS One has a wide-readership, it is critical that the setting and description of the FDMN camps are more fully developed. A reader may be aware of the Rohingya crises, or other humanitarian settings, as I am from the news, but may not have a full understanding of the environment/setting. A fuller description of the services offered and geography is warranted. Understand humanitarian crises.

• Please add more detail to the methods of the study. First, how were study participants selected? Was there a full list and the 34 were selected randomly or selected through another sampling method? Although the background notes 7 camps in the region, were interviews completed in all camps or only 1? How different is the selected camp compared to the others? Additionally, for the analysis, how were discrepancies on themes between the three researchers/coders resolved? A note on the study limitations of the overall methods should be included in the discussions.

• Can the authors differentiate between key informant interview and in depth interview? How did the styles of interview differ? How was it determined who received which type of interview?

• The CFIR framing is extremely helpful to categorize themes and findings. I believe a graphic of the CFIR framework overlaid with themes discussed would be very helpful.

• The results section contains a lot of information, but is very choppy. It would help to have the themes described more broadly for each section and supported with the quotations. For example, if security was the most prominent theme, it would be appropriate to have more than one quote for that section.

Minor:

• Consider converting the maternal and pregnancy related deaths to the global standard maternal mortality ratio (deaths per 100,000) for standard comparison and allow readers to understand the significance of this mortality (line 98-99).

• Please clarify two phrases: (1) Upzila (in line 204) and (2) ‘Majhi’ with addition descriptions. If I understand correctly, the ‘Majhi’ are community health workers from FDMN communities, but based on line 328 it is unclear if they are health trained or just focused on community engagement.

• In the abstract the phrase, ‘our study identified security of the service providers at different level,’ is unclear as to the meaning of the expression, “at different level”? Please revise this sentence in the abstract.

• I note that there are co-first and co-senior authors. I believe the standards in published are that the co-senior authors should be placed together. Please consider moving Dr. Sharkey to be next to last and end with Dr. Shahabuddin.

6. PLOS authors have the option to publish the peer review history of their article (what does this mean?). If published, this will include your full peer review and any attached files.

Reviewer #1: Yes: Md. Fosiul Alam Nizame

Reviewer #2: No

---

## [Author Response · Author response to Decision Letter 0]

1 Mar 2020

Responses to Reviewers

Reviewer #1: Minor Comments:

• You have conducted 15 KIIs and 19 IDIs, but what was the difference between IDI and KII regard to objectives? In your sample table how you have distinguish the IDI and KII participants? Were there different interview guidelines for two different groups? Objective(s) was different? Findings were different? Clarifying these issues could strengthen the article.

o Response: Thank you a lot for your valuable comment. The major difference between the two types of interviews conducted for our study was that; we conducted the KIIs with selected major stakeholders working for the FDMNs in managerial positions or above while, IDIs were with the selected participants, engaged in front line service delivery for the FDMNs who shared their own experience. In the sample table provided at the beginning of the result, section has two separate parts showing who were selected for KIIs and who for IDIs from different GO, INGO, NGO, and UN organization. our interview guides for the KIIs differ from that of the IDIs in a sense that, they had additional questions, suitable for the acquiring information towards the managerial decisions being taken inside the FDMN camps (Line ___)

• Author has mentioned that they conducted 15KIIs and 19 IDIs, but did not mention that why it was 15/19? Data saturation reached or not or any constraints in proceeding for more.

o Response: Thank you again for this question. We continued conducting interviews (KIIs and IDIs) until we reached the saturation point. 

Declaration of competing interests: 'I declare that I have no competing interests'.

Reviewer #2: Thank you for the opportunity to review the paper entitled, “Effective maternal, newborn, and child health programming among Rohingya refugees in Cox’ Bazar, Bangladesh: Implementation challenges and potential solutions.” The authors are to be commended for working on such a revealing and well-documented study in the challenging environment of a humanitarian crisis. The qualitative study of 24 in-depth interviews and key informant interviews provides insight into the challenges and suggested solutions for improving maternal, newborn and child services in the Rohingya camps. I believe this manuscript will add value to the literature around humanitarian and fragile settings. There are some areas for consideration and refinement.

Major:

• Comment: As PLoS One has a wide-readership, it is critical that the setting and description of the FDMN camps are more fully developed. A reader may be aware of the Rohingya crises, or other humanitarian settings, as I am from the news, but may not have a full understanding of the environment/setting. A fuller description of the services offered and geography is warranted. Understand humanitarian crises.

o Response: Thank you very much for the comments, and we also agree with the reviewer. We have included the following text in the manuscript to give the reader a broader overview of the situation, from line number 40 “In total five campsites situated in Ukhia and Teknaf Subdistrict of Cox's Bazar, Ukhia shelters more than two-third of the total refugee population (Figure 01). The refugees living in these settlements are sufferers of hunger, nutrition, safety, and other medical emergencies. The last Joint Response Plan (JRP 2019), identified that more than one million FDMNs were in need of different health-related services [8].”

We have also added the following text from line number 54;

“The FDMNs receive healthcare services from the primary health centers and health posts located within the camps. More than 200 doctors and nurses are currently providing services in the health centers and health posts. These health centers and health posts provide a variety of MNCH services, including Antenatal Care (ANC), Post Natal Care (PNC), referral, normal deliveries, counseling on IYCF indicators, and so on. Apart from the facility-based services, more than 1200 community health workers are providing community-based counseling and other services and assisting in referral to different facilities [11].”

Reference added in serial 11: 

FDMN to Bangladesh: Health Situation & Interventions Update. http://103.247.238.81/webportal/pages/controlroom_rohingya.php (accessed December 13, 2019).

• Comment: Please add more detail to the methods of the study. First, how were study participants selected? Was there a full list, and the 34 were selected randomly or selected through another sampling method?

o Response: We agree with the reviewer and included additional information.

• Line number 90 (in study site).

o The camps in Ukhia alone accommodate 734,645 of the total FDMN population, and currently, 27 organizations, including UNHCR are providing varieties of services in those camps (Figure 02).

• After line number 104;

o We sampled our study participants purposively. We conducted the KIIs with upper mid-level and senior managers working at different levels in different major organizations (GO, NGO, INGO, UN). They are well acquainted with the service, national policies, implementation strategies, and delivery mechanisms. The IDIs respondents were health service providers working for relevant organizations and responsible for providing health services to the camps. In total, we conducted 34 interviews (15 KIIs and 19 IDIs). We asked the participants about MNCH challenges and potential solutions circumnavigating the implementation of the program. We contacted the participants beforehand to ensure their availability, with the help of Unicef Cox's Bazar, our local partner, for the study. We pre-tested the IDI & KII guidelines in a similar facility, which was not part of the primary research before the data collection. A seven-member research team experienced in conducting qualitative interviews took part as the interviewers, all of whom had received training before going to the field to get familiar with the guideline and study objectives. We carried out the interview sessions in a setting free from outside interference so that the study participants could feel at ease in sharing their views and perceptions. We conducted our interviews until there was no new information generating from the interviews. 

• Comment: Although the background notes seven camps in the region, were interviews completed in all camps or only 1? How different is the selected camp compared to the others?

o Response: The seven camps are apparently homogeneous. The KIIs were conducted with the major stakeholders (Government, NGO, INGO, UN) working on all campuses. It included midlevel and senior managers, including policymakers of the different national, international, and UN organizations, and the Government officials working in all seven camps. However, for IDI (service providers), we have selected the largest camp 'Kutupalong" to capture the experience of the continuum of MNCH care because the primary care to referral center is located in "Kutupalong".

• Comment: Additionally, for the analysis, how were discrepancies on themes between the three researchers/coders resolved?

o Response: Thank you for the comment. We have added the following sentence from line number 141;

“To avoid any sort of intra-code discrepancy among the researchers who took part in the analysis, we cross-checked and counter-checked all the codes that emerged and organized the themes when we reached a consensus.”

• Comment: A note on the study limitations of the overall methods should be included in the discussions.

o Response: We have included the limitation in the discussion from line number 458.

• Comment: Can the authors differentiate between key informant interview and in depth interview? How did the styles of interview differ? How was it determined who received which type of interview?

o Response: Thank you a lot for your valuable comment. The major difference between the two types of interviews conducted for our study was that; we conducted the KIIs with selected major stakeholders working for the FDMNs in managerial positions or above while, IDIs were with the selected participants, engaged in front line service delivery for the FDMNs who shared their own experience. In the sample table, the beginning of the result section has two separate parts showing who were selected for KIIs and who for IDIs from different GO, INGO, NGO, and UN organizations. Our interview guides for the KIIs differ from that of the IDIs in a sense that, they had additional questions, suitable for the acquiring information towards the managerial decisions being taken inside the FDMN camps.

• Comment: The CFIR framing is extremely helpful to categorize themes and findings. I believe a graphic of the CFIR framework overlaid with themes discussed would be very helpful.

o Response: Thank you for the valuable feedback, and we agree with the reviewer. Hence, we have included a figure titled- "CFIR constructs and overlaid themes with associated challenges and facilitators” as Figure 03 (line number 146). 

• Comment: The results section contains a lot of information, but it is very choppy. It would help to have the themes described more broadly for each section and supported with the quotations. For example, if security was the most prominent theme, it would be appropriate to have more than one quote for that section.

o Response: Thank you for the comment. However, we disagree. This is the only humanitarian setting in Bangladesh and the first implementation research on MNCH services. The information related to MNCH is very important for the implementors and policymakers to change the implementation strategy or program activities. Making it too broad will dilute the granules of the program information, which is not useful for the audiences.

Minor:

• Comment: Consider converting the maternal and pregnancy-related deaths to the global standard maternal mortality ratio (deaths per 100,000) for standard comparison and allow readers to understand the significance of this mortality (line 98-99).

o Response: We agree with the reviewer. However, the denominator (live birth) data is partially available as many births are taking place at home, and no systematic live birth data collection is in place. Therefore, maternal mortality ration cannot be calculated. Hence we have kept the sentence in line number 67-68 unchanged.

• Comment: Please clarify two phrases: (1) Upazila (inline 204) and (2) 'Majhi' with addition descriptions. If I understand correctly, the 'Majhi' are community health workers from FDMN communities. Still, based on line 328, it is unclear if they are health trained or just focused on community engagement. 

o Response: We agree with the reviewer. Here, 'Upzila' is the Bengali name of 'Sub-district', and this might create confusion among different readers. Hence, we have changed it to ‘Sub-district level Health facility’ in line number 189. As for the 'Majhi'; they are not community health workers as they are not paid by anyone, rather they act more like community-level volunteers who help other CHWs in their community mobilization works by using their influence as the local community leaders. And they are appointed by the organizations, not the CHWs either. Hence, in line number 314, we have rephrased it as follow;

“The 'Majhi' is a community leader in the FDMN population and appointed by the different organizations to act as community-level volunteers for the social mobilization activities, often helps other CHWs in their works by using their influence as the local community leaders.”

• Comment: In the abstract, the phrase, 'our study identified security of the service providers at a different level,' is unclear as to the meaning of the expression "at different level"? Please revise this sentence in the abstract.

o Response: We agree with the reviewer, and we have rephrased the specified sentence as follow;

 "Our study identified some major challenges hindering the effective implementation of MNCH service delivery for the FDMNs. Providing a sense of security for the service providers, high turnover and poor retention of staff, overlapping of service, weak referral mechanism, complex health information system, and so on were some of the key challenges identified."

• Comment: I note that there are co-first and co-senior authors. I believe the standards in published are that the co-senior authors should be placed together. Please consider moving Dr. Sharkey to be next to last and end with Dr. Shahabuddin.

o Response: Thank you for your comment. However, according to the author's guideline from PLoS One, the author's order is based on the contribution in manuscript writing. Specifically, the eligibility of senior authorship is overall guidance in the project. Dr. ASM Shahabuddin is the only eligible senior author.

Additional edit made:

1. Dr. ASM Shahabuddin’s affiliation had a mistake. Hence, we have corrected it in the author list.

---

## [Editor Report · Decision Letter 1]

9 Mar 2020

Effective maternal, newborn and child health programming among Rohingya refugees in Cox’s Bazar, Bangladesh: Implementation challenges and potential solutions

PONE-D-19-28648R1

Dear Dr. Sarker,

We are pleased to inform you that your manuscript has been judged scientifically suitable for publication and will be formally accepted for publication once it complies with all outstanding technical requirements.

With kind regards,

Vijayaprasad Gopichandran

Academic Editor

PLOS ONE
---

## [Editor Report · Acceptance letter]

13 Mar 2020

PONE-D-19-28648R1 

Effective maternal, newborn and child health programming among Rohingya refugees in Cox’s Bazar, Bangladesh: Implementation challenges and potential solutions 

Dear Dr. Sarker:

I am pleased to inform you that your manuscript has been deemed suitable for publication in PLOS ONE. Congratulations! Your manuscript is now with our production department. 

With kind regards,

on behalf of

Dr. Vijayaprasad Gopichandran 

Academic Editor

PLOS ONE